# Chemo-, regio- and stereoselective access to polysubstituted 1,3-dienes via Nickel-catalyzed four-component reactions

Shanglin Chen[1,5], Ya-Nan Wang[2,5], Jinhui Xie[1], Wangyang Li [1], Mingxing Ye[1], Xingxing Ma[1], Kai Yang [1], Shijun Li [3], Yu Lan [2,3] ✉ & Qiuling Song [1,4] ✉

1,2-Difunctionalization of alkynes offers a straightforward approach to access polysubstituted alkenes. However, simultaneous multi-component cascade transformations including difunctionalization of two alkynes with both syn- and anti-selectivity in one catalyst system is undeveloped and proves to be a significant challenge. Herein, we report a Nickel-catalyzed four-component reaction to access polysubstituted 1,3-dienes using two terminal alkynes, aryl boroxines, and perfluoroalkyl iodides, wherein the reaction forms three new C-C bonds in a single vessel and serve as a modular strategy to access polysubstituted 1,3-dienes with excellent chemoselectivity, good regioselectivity and exclusive stereoselectivity. Control experiments reveal the plausible reaction mechanism and DFT calculations explain the cause for the formation of this unusual four-component reaction. Furthermore, we successfully incorporate two biologically active units into 1,2,3,4-tetrasubstituted 1,3-dienes, which greatly increases the diversity of molecular scaffolds and brings more potential values to medicinal chemistry, the synthetic utility of our protocol is further demonstrated by the late-stage transformations.

Transition-metal-catalyzed 1,2-difunctionalization of alkynes represents an intriguing and prevailing tactic for the efficient construction of diverse valuable polysubstituted alkenes[1–10] and their synthetic applications widely extend to pharmaceuticals, agrochemicals, natural products as well as material sciences[11–15], thus stimulating great attention in the chemical community. In the past decade, the development of various coupling partners greatly increases organic molecular complexity and facilitates reaction diversification[13,16,17]. Among them, organoboron reagents are attractive and valuable synthons owing to their air stability, non-toxicity, abundance, readily accessible, and broad availability in versatile transformations[13,18–20]. Hence, transition-metal-catalyzed 1,2-difunctionalization of alkynes involving organoboron reagents[13,17,21] has been proven to be one of the most versatile

and powerful tools for the efficient construction of diverse chemical bonds recently. Based on the mechanism, this type of transformation could be divided into radical addition/cross-coupling tactic and carbometallation/cross-coupling platform[3,5,16], and both of them are known strategies for the synthesis of value densely substituted alkenes (Fig. 1a). On the one hand, migratory insertion of alkyne with organometallic species generating from organoboron reagents enables the carbometallation procedure, forming alkenyl-metal intermediates which are further trapped by external electrophiles[13], thus affording the syn-difunctionalization of alkynes, and the successful transition-metal catalysis[3,5,13] which could promote such reactions includes Ni[22], Rh[23,24], Pd[25,26], etc. (Fig. 1a, left). In this process, the use of directing group-containing[5,13,25–28] or electronically biased alkynes[5,13,29–31] have

[1]Key Laboratory of Molecule Synthesis and Function Discovery, Fujian Province University, College of Chemistry at Fuzhou University, Fuzhou, Fujian, China. [2]Chongqing Key Laboratory of Theoretical and Computational Chemistry, School of Chemistry and Chemical Engineering, Chongqing University, Chongqing, China. [3]College of Chemistry and Institute of Green Catalysis, Zhengzhou University, Zhengzhou, China. [4]School of Chemistry and Chemical Engineering, Henan Normal University, Xinxiang, Henan, China. [5]These authors contributed equally: Shanglin Chen, Ya-Nan Wang. ✉e-mail: ylan@zzu.edu.cn; qsong@fzu.edu.cn

been critical to control regioselectivity, yet the success of syn-difunctionalization of accessible unsymmetrical alkynes with complete regioselectivity is full of challenges. On the other hand, on account of steric factors of vinyl radicals in the coupling step, radical addition/cross-coupling platform favors anti-addition[3,5,13], which typically transforms into a stereodefined anti-difunctionalization of alkynes by utilizing the commercially available organoboron compounds as nucleophilic reagents[32–39], and recent feasible catalysts for this transformation are Pd[32–36], Ni[37,38], Cu[39] etc. (Fig. 1a, right). However, the combination of feasible alkenyl radicals with a more reactive intermediate generated in a reaction system instead of well-studied nucleophilic reagents under transition-metal-catalysis is scarce, and further exploration is still needed. Herein, it should be noted that the above-mentioned three-component difunctionalization protocols were generally used for the assembly of polysubstituted alkenes with syn- or anti-selectivity respectively, which limit to install two units into one alkyne to render the corresponding product, yet the integration of two alkynes with two accessible coupling reagents in a single vessel to deliver value-added polysubstituted 1,3-dienes with excellent stereospecificity in one catalytic system via a four-component reaction has not been disclosed, thus remaining a vast challenge.

Given the rapid development of the combination of radical chemistry with transition-metal-catalyzed transformations[15], we envisioned that the carbometallation platform should be possibly combined with radical-promoted platform to render a four-component process, in details, alkyne undergoes migratory insertion into nucleophilic aryl-metal species with syn-selectivity to lead to the corresponding alkenyl-metal intermediate, concurrently, radical-mediated platform enables radical addition of perfluoroalkyl halides to another alkyne with anti-selectivity to generate a very active vinyl radical, the corresponding alkenyl-metal intermediate could trap the above-mentioned newly generated alkenyl radical intermediate followed by reductive elimination to facilitate cascade four-component reactions (Fig. 1b).

Seeking to achieve the above-mentioned hypothesis about the cascade difunctionalization of two alkynes, we need to face several inherent challenges (Fig. 1b): (1) How to make a breakthrough for conventional three-component difunctionalization reaction from similar accessible starting materials[34]? Intermolecular multi-component reaction processes using similar starting materials are more challenging inherently, as the radical-mediated three-component background reaction acting as the competing reaction greatly hampers the success of our design. At the same time, on account of massive competitive reactions that exist in current multi-component reactions (Such as well-studied three-component reaction, Atom Transfer Radical Addition reaction (ATRA), Radical addition reaction (RA), and Alkyne Insertion reaction (AI)), the capability to efficiently suppress the potential competitive reactions and achieve the specificity of our expected multi-component reaction is full of hurdles. (2) Selective control has always been a formidable challenge in multi-component reactions[40–52]. There is only one regioselectivity involved in reported three-component difunctionalization[32–39] of alkynes, yet in our hypothesis, we pursue to simultaneously enable syn-difunctionalization as well as anti-difunctionalization of alkynes in one catalytic system to afford both syn-controlled alkene unit and anti-controlled alkene unit, therefore, how to precisely regulate the chemo- and regioselectivity of two alkynes in a single pot without the directing groups[13,25–28] and the use of electronically biased[13,29–31] alkynes will be a great challenge. (3) How to afford the expected polysubstituted 1,3-dienes with challenging and intriguing stereospecificity? If successful, this strategy would achieve a four-component reaction with cascade 1,2-functionalization of two alkynes in one catalyst system, make a breakthrough for a well-developed three-component difunctionalization reaction from accessible starting materials[34] and offer an

opportunity to lead to the modular assembly of value-added, appealing as well as structurally diversified fluorinated polysubstituted 1,3-dienes.

1,2,3,4-Tetrasubstituted 1,3-dienes, one of the most complicated and traditionally not-easy-to-accessible conjugated dienes, are widely found in natural products due to their unique biological activity[53] and possess unique structural complexity as well as a great diversity of stereoisomers (include (1E, 3E), (1E, 3Z), (1Z, 3Z), (1Z, 3E)), to the best of our knowledge, the synthesis of such polysubstituted substituted 1,3-dienes with excellent regioselectivity and exclusive stereoselectivity has been limited[53–59] so far and still remains a formidable challenge. Therefore, the exploration of a simple and practical method for the preparation of 1,2,3,4-tetrasubstituted conjugate dienes is highly desired[58]. Herein, we report an Nickel-catalyzed four-component reaction via chemo-, regio-, stereoselective cascade difunctionalizations of two alkynes to access polysubstituted 1,3-dienes (Fig. 1c), which includes simultaneous syn-difunctionalization and anti-difunctionalization of alkynes in one catalytic system, gratifyingly, this strategy also provides an opportunity to build structurally diverse 1,2,3,4-tetrasubstituted 1,3-dienes by utilizing the readily accessible starting materials and cheap transition-metal catalyst in a single operation. Moreover, this reaction features mild reaction conditions, good functional group tolerance, excellent chemoselectivity, good regioselectivity as well as exclusive stereoselectivity.

## Results
### Reaction conditions optimization
To validate our initial hypothesis and solve the aforementioned challenges, we began our research by using methyl 4-ethynylbenzoate **1**, perfluorobutyl iodide **2,** and 4-methoxyaryl boroxine **3** as the model substrates. We guess that this unusual, appealing, and more challenging multi-component intermolecular cross-coupling reaction could be achieved using the proper catalytic system, as the palladium catalysts show excellent performance in previous radical-mediated multi-component difunctionalization reactions[32–36], thus promoting us to have an attempt to utilize them at first. To our delight, when we used 4 mol% Pd(PPh$_3$)Cl$_2$ or Pd(PPh$_3$)$_4$ as the metal catalyst (Table 1, entries 1–4), the polysubstituted 1,3-diene **5** could be successfully obtained with good regioselectivity and exclusive stereoselectivity albeit in poor yield, indicating that we indeed enabled cascade difunctionalization of alkynes in one catalytic system. It was worth noting that polysubstituted alkene **4** was the dominant byproduct in this reaction condition, which greatly affected the success of our designed reaction. After extensive and time-consuming condition evaluations regarding palladium catalysts (Please see Supplementary Table 1 for more details), no satisfactory outcomes could be obtained, and this reaction tendency was exclusively dominated by conventional three-component reaction or simply afforded the desired product **5** in poor yields, which forced us to examine different metal catalyst systems. In order to realize our initial assumption, Fe[60] and Cu[61,62] catalysts were further assessed since they also showed good performance in the difunctionalization of alkynes in previous reports, (Table 1, entries 5–15), however, neither polysubstituted alkenes **4** nor polysubstituted 1,3-dienes **5** were detected in the system, suggesting that these metal catalysts were not compatible with our designed reaction. Nonetheless, the above unsatisfactory results did not discourage us, and we subsequently focused on a Nickel-catalyzed system based on the known reactions[37,38]. We pleasingly found that the nitrogen ligands performed better than the phosphine ligands in the Nickel-catalyzed reaction system (Table 1, entries 16–21), moreover, polysubstituted 1,3-diene **5** began to become the primary product and only a trace amount of polysubstituted alkene **4** was obtained, this result revealed that the nickel catalytic system is more favorable for four-component reaction instead of classical three-component reaction, thus promoted us to

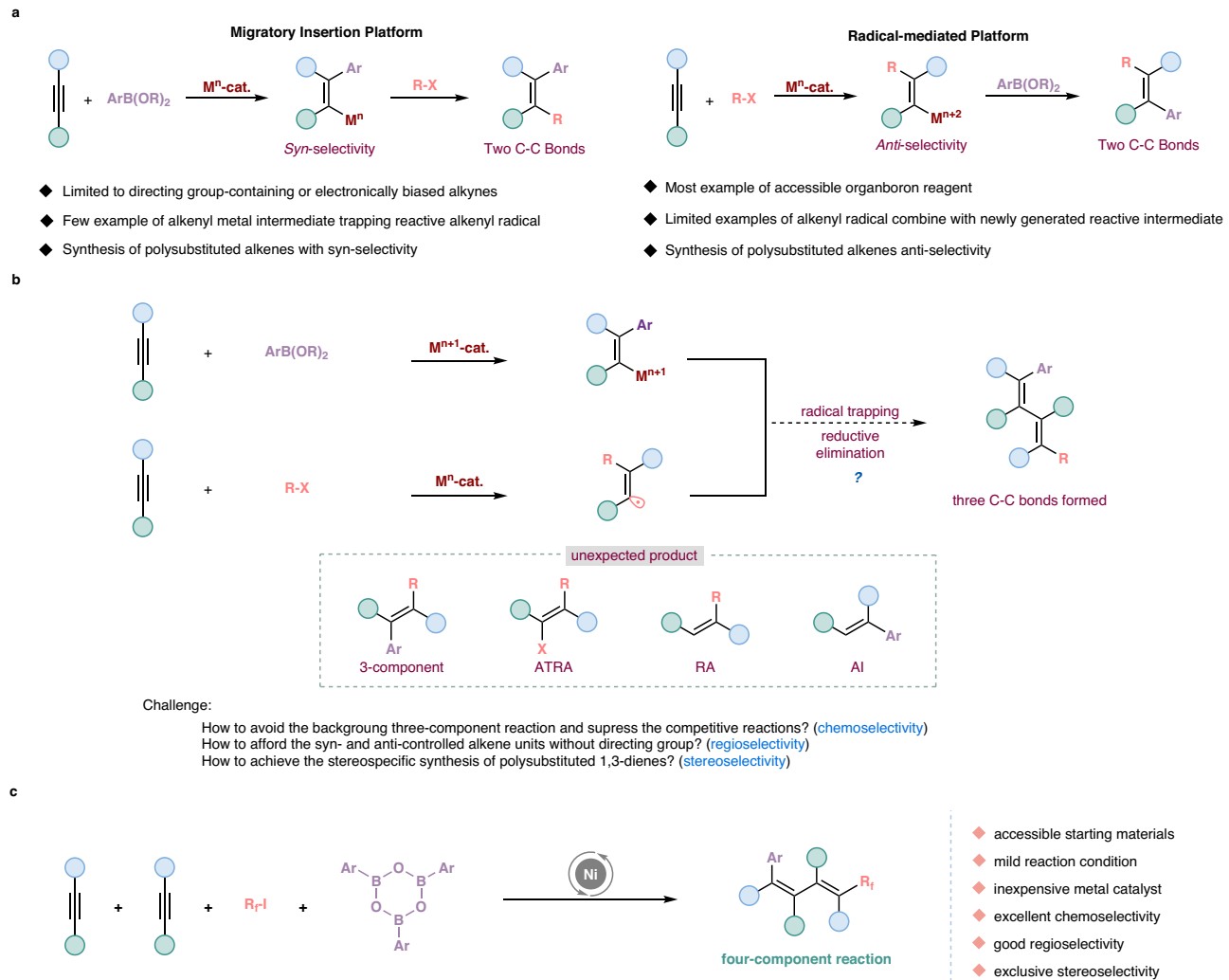

**Fig. 1 | Background and our strategy. a** Previous arts on transition-metal-catalyzed 1,2-difunctionalization of alkynes (three-component reaction, well-studied). **b** Our design (four-component reaction, unknown). **c** Synthesis of poly-substituted 1,3-dienes via nickel-catalyzed four-component reaction with two alkynes (**this work**). 3-component = three-component reaction, ATRA = atom transfer radical addition reaction, RA = radical addition reaction, AI = alkyne insertion reaction.

utilize nickel catalysts to optimize the reaction conditions for this unusual cascade four-component reaction.

Subsequently, by evaluation of metal catalysts and ligands, we found that $Ni(PCy_3)_2Cl_2$ and dtbpy emerged as lead catalysts and superior ligands in our catalytic system (Table 2, entries 1–7; please see Supplementary Table 5 and Table 6 for more details). Notably, no desired 1,3-diene **5** was detected when we used $Cs_2CO_3$[32,33] or $K_3PO_4$[36,37] as the base (Table 2, entries 8-9), which could effectively promote nickel-catalyzed or palladium-catalyzed three-component 1,2-difunctionalization of alkynes involving organoboron reagent in previous works. Upon the replacement of aryl boroxine with the corresponding potassium trifluoroborate or pinacol ester, the desired product **5** could not be obtained (Table 2, entries 10, 11), to our surprise, aryl boronic acid could get relatively good results in 60 °C (Table 2, entry 12), while it still remained room for further advancement. Pleasingly, using DMA and DME as mixed solvents to our reaction system, outperformed the single solvent system (Table 2, entries 13, 14). This phenomenon definitely prompted us to investigate more mixed solvent systems and diverse proportions of mixed solvent, yet it was regretful that no better outcomes were achieved in these experiments (Please see the Supplementary Table 7 for more details). Encouraged by the above results, we further carefully screened the temperatures and concentrations

(Table 2, entries 15, 16, see the Supplementary Table 9 and Table 12 for more details), and eventually the optimal condition for this reaction was listed as entry 2 with 71% isolated yield, additionally, when the reaction was scaled up to 1 mmol scale, the target product **5** was obtained in decent yield without significant erosion of the efficiency (Table 2, entry 2). By comparison, a dramatic decrease was observed when the reaction was performed in the absence of a nickel catalyst, ligand, or base (Table 2, entries 18–20), further indicating that they are essential to carry out this multi-component reaction system. In particular, it should be noted that only a small amount of byproduct polysubstituted alkene **4** was detected in this whole process, thus revealing that the Ni-catalyzed system was indeed suitable catalytic system for our designed reaction and we successfully modulated the reaction from conventional three-component reaction transform into unusual four-component reaction under mild condition.

## Substrate scopes

With the optimal reaction condition in hand (Table 2, entry 2), we next implemented to examine the substrate scope of this cascade four-component reaction (Fig. 2). At the outset, we explored the substrate with different substituents on the para position of the aromatic alkynes. To our delight, no matter that they were hydrogen, methyl,

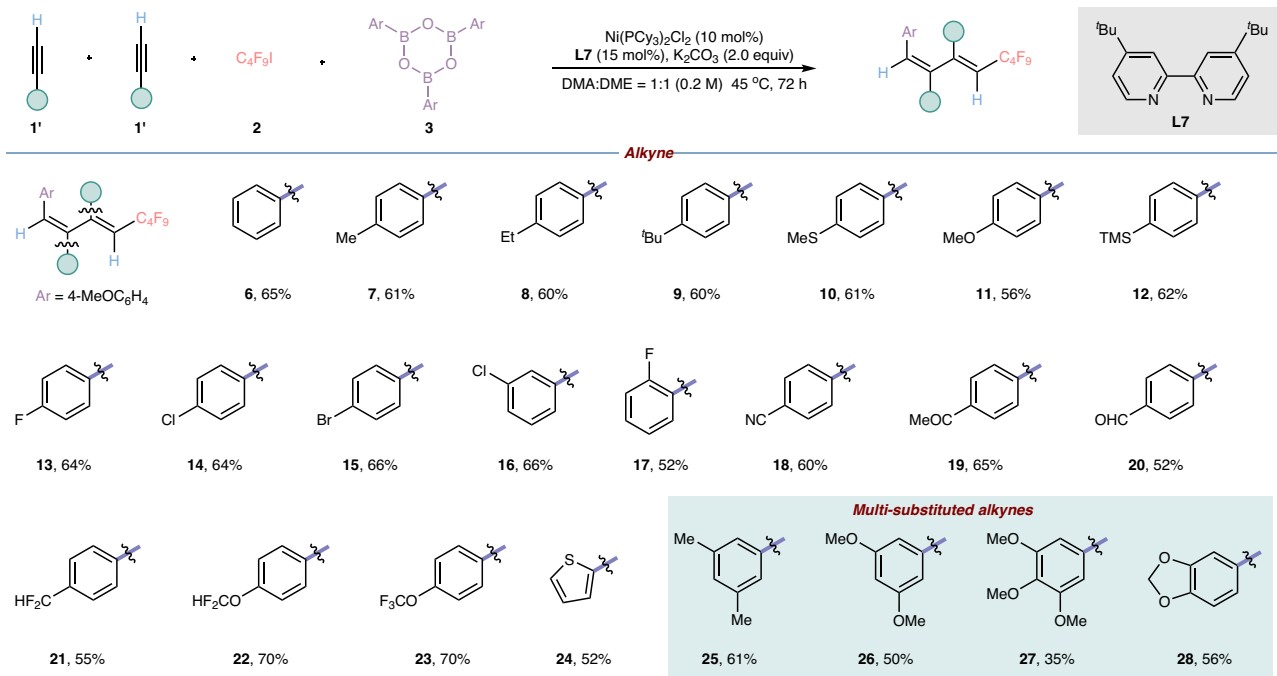

**Fig. 2 | Substrate scope of catalytic cascade difunctionalizations of alkynes.** Reaction conditions: the reaction was carried out with alkyne **1'** (0.27 mmol), **2** (0.8 mmol), **3** (0.067 mmol), Ni(PCy₃)₂Cl₂ (10 mol%), dtbpy (15 mol%), K₂CO₃ (0.4 mmol, 2.0 equiv) in the 1 mL of mixed solvent (DMA:DME = 1:1) at 45 °C for 72 h. Unless otherwise stated, all desired polysubstituted 1,3-dienes were obtained with regioselectivity greater than 95:5 and the regioselectivity was detected by ¹H NMR analysis of desired products or GC analysis of desired products. DMA = N, N-dimethylacetamide, DME = 1,2-dimethoxyethane.

ethyl, tert-butyl, methylthio, methoxy, and trimethylsilyl groups, the corresponding products (**6–12**) were all smoothly delivered in moderate yields with excellent regioselectivity and exclusive stereoselectivity. Then we evaluated the different halogen-substituted aromatic alkynes and found that they were compatible to our reaction, as well in spite of their positions on the aromatic rings (**13–17**). Meanwhile, the general electron-withdrawing groups such as ester, cyano, ketone, aldehyde, difluoromethyl, difluoromethoxy as well as trifluoromethoxy group were all tolerated with our system and the target products (**5**) and (**18–23**) were obtained in good outcomes. Heteroaromatic and polysubstituted alkynes including thiophene-substituted, disubstituted and trisubstituted alkynes (**24–28**) were also suitable substrate for our multi-component reaction and worked smoothly and delivered the desired polysubstituted 1,3-dienes in moderate yields. Unfortunately, alkyl-substituted alkyne and 1,3-enyne were not suitable substrates for our current reaction system to access expected polysubstituted 1,3-dienes, and two molecules of different terminal alkynes were also not compatible with our current strategy owing to poor chemoselectivity of radical addition process and migratory insertion process (Please see the Supplementary 7.4 for more details).

Encouraged by the above results, we next explored the scope of aryl boroxines under our optimal reaction condition, and the results were summarized in Fig. 3. We turned our attention to the reaction of various aryl boroxine **3'** with methyl 4-ethynylbenzoate **1** and perfluorobutyl iodide **2**. It was worth noting that the poor outcomes with certain aryl boroxines resulting from the standard reaction conditions could be rectified by elevating the reaction temperature from 45 °C to 50 °C (**29, 33, 43-44, 54-59**). To our delight, this change not only prompted the reaction to proceed smoothly, but also maintained the excellent regioselectivity and exclusive stereoselectivity of the expected polysubstituted 1,3-dienes. It was found that this cascade difunctionalization reaction exhibited good functional group tolerance, a wide range of the para-, meta- and ortho-substituents on aryl boroxines were examined, both electron-neutral and electron-

donating groups, demonstrated good compatibility, and the correspondingly desired products were obtained in moderate to good yields with excellent chemoselectivity, exclusive regio- and stereoselectivity (**29–40**), upon treatment with our standard reaction condition. Furthermore, general halogen-substituted aryl boroxines were also good candidates in our reaction and rendered the desired halogen-substituted 1,2,3,4-tetrasubstituted dienes (**42–45**), which could be used for further elaborations. Gratifyingly, the success of this four-component reaction could be extended to alkyl boroxine as well and the desired polysubstituted 1,3-diene **41** was procured in moderate yield. Moreover, this four-component reaction was applicable to diverse (hetero) aryl boroxines and polysubstituted aryl boroxines, which successfully transform into our desired products (**46–53**) in moderate to good yields. A diminished yield was observed for ortho-substituted aryl boroxine (**40**), which might be attribute to steric hindrance.

To further demonstrate the generality of this cascade four-component reaction, we examined the substrate scope of radical precursors (Fig. 3). Concretely, the 1,2,3,4-tertasubstituted 1,3-dienes (**54–59**) could be obtained in moderate yields with excellent regioselectivity and exclusive stereoselectivity by increasing the reaction temperature to 50 °C. The absolute structures of products **58** and **59** were determined by X-ray crystallographic analysis, which revealed that we indeed achieve the cascade syn- and anti-difunctionalization of two terminal alkynes in one catalytic system and successfully access the structurally diverse 1,2,3,4-tertasubstituted conjugated 1,3-dienes from readily accessible starting materials, inexpensive metal catalyst and available nitrogen ligand in a single step under mild condition. Notably, 1,4-diiodoperfluorobutane which possesses reaction sites was also a good substrate for our system reaction and the desired polysubstituted 1,3-diene **57** was afforded in moderate yield. Furthermore, we also examined a variety of non-fluoroalkyl radical precursors in our current catalytic system, yet they were unable to afford the desired products (Please see Supplementary 7.5 for more details).

**Table 1 | Initial attempt and optimization conditions regarding metal catalysts and ligands**

| entry | [M] | ligand | base | 4 yield (%)[e] | 5 yield (%)[e] |
|---|---|---|---|---|---|
| 1[a] | Pd(PPh₃)₂Cl₂ | - | K₂CO₃ | 71 | 26 |
| 2[a] | Pd(PPh₃)₄ | - | K₂CO₃ | 72 | 29 |
| 3[a] | Pd(PCy₃)₂Cl₂ | - | K₂CO₃ | 14 | 14 |
| 4[a] | Pd(CN)₂Cl₂ | - | K₂CO₃ | 6 | 21 |
| 5[b] | FeBr₂ | - | Cs₂CO₃ | nd | nd |
| 6[b] | FeCl₂ | - | Cs₂CO₃ | nd | nd |
| 7[b] | Fe(acac)₃ | - | Cs₂CO₃ | nd | nd |
| 8[b] | Fe(OTf)₂ | - | Cs₂CO₃ | nd | nd |
| 9[b] | Fe(OAc)₂ | - | Cs₂CO₃ | nd | nd |
| 10[c] | CuI or Cu(CH₃CN)₄BF₆ | L1 | K₂CO₃ | nd | nd |
| 11[c] | CuI or Cu(CH₃CN)₄BF₆ | L2 | K₂CO₃ | nd | nd |
| 12[c] | CuI or Cu(CH₃CN)₄BF₆ | L3 | K₂CO₃ | nd | nd |
| 13[c] | CuI or Cu(CH₃CN)₄BF₆ | L4 | K₂CO₃ | nd | nd |
| 14[c] | CuI or Cu(CH₃CN)₄BF₆ | L5 | K₂CO₃ | nd | nd |
| 15[c] | CuI or Cu(CH₃CN)₄BF₆ | L6 | K₂CO₃ | nd | nd |
| 16[d] | NiBr₂·DME | L1 | K₂CO₃ | 5 | 36 |
| 17[d] | NiBr₂·DME | L2 | K₂CO₃ | 8 | 3 |
| 18[d] | NiBr₂·DME | L3 | K₂CO₃ | 8 | 32 |
| 19[d] | NiBr₂·DME | L4 | K₂CO₃ | nd | nd |
| 20[d] | NiBr₂·DME | L5 | K₂CO₃ | nd | nd |

[a]Reaction conditions: the reaction was carried out with **1** (0.25 mmol), **2** (0.8 mmol), **3** (0.067 mmol), palladium catalyst (4 mol%), K₂CO₃ (2.0 equiv) in 2 mL of mixed solvent (DCM:H₂O = 5:1) at 50 °C for 12 h. [b]Reaction conditions: the reaction was carried out with **1** (0.25 mmol), **2** (0.8 mmol), **3** (0.067 mmol), iron catalyst (10 mol%), Cs₂CO₃ (1.5 equiv) in 2 mL 1,4-dioxane at 80 °C for 12 h. [c]Reaction conditions: the reaction was carried out with **1** (0.25 mmol), **2** (0.8 mmol), **3** (0.067 mmol), copper catalyst (10 mol%), ligand (10 mol%), K₂CO₃ (3.0 equiv) in 2 mL toluene at 80 °C for 12 h. [d]Reaction conditions: the reaction was carried out with **1** (0.25 mmol), **2** (0.8 mmol), **3** (0.067 mmol), nickel catalyst (10 mol%), ligand (15 mol%), K₂CO₃ (2.0 equiv) in 2 mL DMA at 70 °C for 24 h. [e]yields were determined by gas chromatography (GC) using n-dodecane as the internal standard. Unless otherwise stated, the desired polysubstituted 1,3-dienes were obtained with regioselectivity greater than 95:5 (the ratio of **5:5'** was greater than 95:5), and the regioselectivity was detected by ¹H NMR analysis of desired products or GC analysis of desired products. M = Metal catalysts, L = Ligand, nd =not detected.

**Table 2 | Reaction condition optimizations regarding nickel-catalyzed system.** [a]

| entry | [Ni] | ligand | base | solvent | 5 yield (%)[b] |
|---|---|---|---|---|---|
| 1 | Ni(PCy₃)₂Cl₂ | L1 | K₂CO₃ | DMA: DME (1:1) | 71 |
| 2 | Ni(PCy₃)₂Cl₂ | L7 | K₂CO₃ | DMA: DME (1:1) | 81(71)[c](65)[d] |
| 3 | Ni(PCy₃)₂Cl₂ | L8 | K₂CO₃ | DMA: DME (1:1) | 68 |
| 4 | Ni(PCy₃)₂Cl₂ | L9 | K₂CO₃ | DMA: DME (1:1) | 70 |
| 5 | Ni(PCy₃)₂Cl₂ | L10 | K₂CO₃ | DMA: DME (1:1) | trace |
| 6 | Ni(PCy₃)₂Cl₂ | L11 | K₂CO₃ | DMA: DME (1:1) | trace |
| 7 | Ni(PCy₃)₂Br₂ | L7 | K₂CO₃ | DMA: DME (1:1) | 51 |
| 8 | Ni(PCy₃)₂Cl₂ | L7 | Cs₂CO₃ | DMA: DME (1:1) | nd |
| 9 | Ni(PCy₃)₂Cl₂ | L7 | K₃PO₄ | DMA: DME (1:1) | nd |
| 10 | Ni(PCy₃)₂Cl₂ | L7 | K₂CO₃ | DMA: DME (1:1) | nd[e] |
| 11 | Ni(PCy₃)₂Cl₂ | L7 | K₂CO₃ | DMA: DME (1:1) | trace[f] |
| 12 | Ni(PCy₃)₂Cl₂ | L7 | K₂CO₃ | DMA: DME (1:1) | 63[g] |
| 13 | Ni(PCy₃)₂Cl₂ | L7 | K₂CO₃ | DMA | 56 |
| 14 | Ni(PCy₃)₂Cl₂ | L7 | K₂CO₃ | DME | 25 |
| 15 | Ni(PCy₃)₂Cl₂ | L7 | K₂CO₃ | DMA: DME (1:1) | 71[h] |
| 16 | Ni(PCy₃)₂Cl₂ | L7 | K₂CO₃ | DMA: DME (1:1) | 68[i] |
| 17 | Ni(PCy₃)₂Cl₂ | L7 | K₂CO₃ | DMA: DME (1:1) | 76[j] |
| 18 | - | L7 | K₂CO₃ | DMA: DME (1:1) | nd |
| 19 | Ni(PCy₃)₂Cl₂ | - | K₂CO₃ | DMA: DME (1:1) | <5 |
| 20 | Ni(PCy₃)₂Cl₂ | L7 | - | DMA: DME (1:1) | nd |

[a]The reaction was carried out with **1** (0.27 mmol), **2** (0.8 mmol), **3** (0.067 mmol), Ni catalyst (10 mol%), ligand (15 mol%), K₂CO₃ (2.0 equiv) in 1 mL of mixed solvent (DMA: DME = 1:1) at 45 °C for 72 h. [b]Yields were determined by gas chromatography (GC) using n-dodecane as the internal standard. [c]Isolated yield. [d]1 mmol scale reaction. [e]ArB(OH)₂ was used. [f]ArBpin was used. [g]ArBF₃K was used. [h]50 °C. [i]40 °C. [j]2 mL of mixed solvent (DMA: DME = 1:1). Unless otherwise stated, the desired polysubstituted 1,3-dienes were obtained with regioselectivity greater than 95:5 (the ratio of **5/5'** was greater than 95:5) and (1Z, 3E)/(1E, 3Z) of desired polysubstituted 1,3-dienes were greater than 95:5, and the ratio of Z/E and regioselectivity were detected by ¹H NMR analysis of desired products or GC analysis of desired products, the polysubstituted alkene **4** was obtained with the yield lower than 10%. nd, not detected.

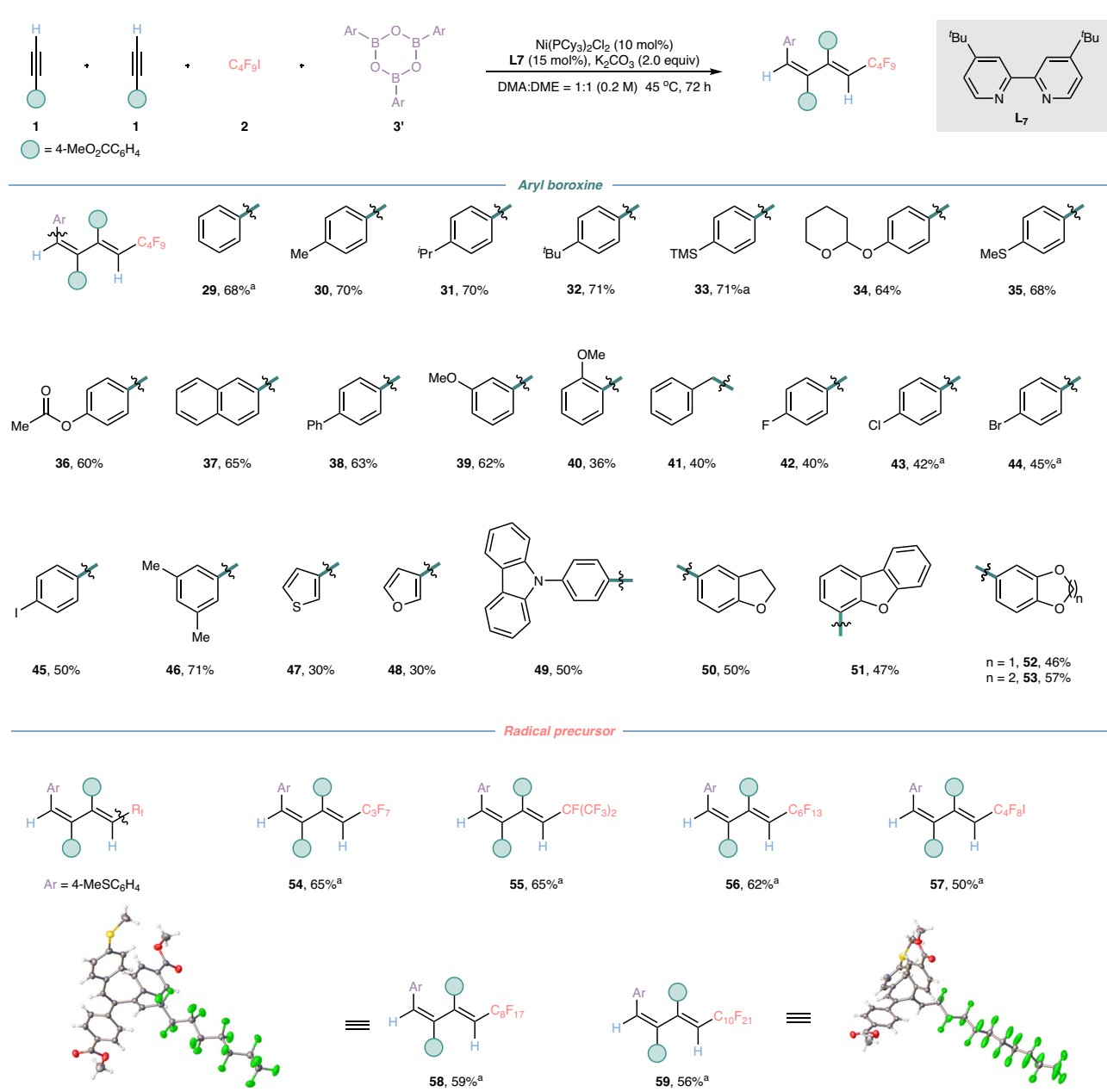

**Fig. 3 | Substrate scope of aryl boroxines and radical precursors.** Reaction conditions: the reaction was carried out terminal alkyne (0.27 mmol), perfluorobutyl iodine or perfluoroalkyl iodide (0.8 mmol), aryl boroxines (0.067 mmol), Ni(PCy$_3$)$_2$Cl$_2$ (10 mol%), dtbpy (15 mol%), K$_2$CO$_3$ (2.0 equiv) in 1 mL of mixed solvent (DMA:DME = 1:1) at 45 °C for 72 h. Unless otherwise stated, all desired polysubstituted 1,3-dienes were obtained with regioselectivity greater than 95:5; and the regioselectivity was detected by $^1$H NMR analysis of desired products or GC analysis of desired products. [a]The reaction was carried out in 50 °C. DMA = N, N-dimethylacetamide, DME = 1,2-dimethoxyethane.

To further showcase the high functional group compatibility and broad substrate scope of our current transformation, we applied this strategy to the late-stage functionalization of bioactive compounds and drug molecules (Fig. 4). It is noteworthy that this multi-component reaction performed well with aromatic alkynes derived from Borneol, Adamantanemethanol, Galactopyranose, Gemfibrozil, L-Menthol, Ibuprofen and Cholesterol (60–66), affording the expected polysubstituted 1,3-dienes bearing two complex molecular scaffolds in moderate yields with excellent regioselectivity and exclusive stereoselectivity, these scarce structures probably bring more potential values to medicinal chemistry. In addition, the carbon-carbon double bond was also compatible with our reaction system (66), thus further confirming the good compatibility of this four-component reaction.

**Downstream applications and transformations**

To exemplify the synthetic utility of the current method, a series of transformations on compound **45** were performed in Fig. 5. At the outset, we tried to convert the reactive C-I bond of polysubstituted 1,3-dienes which derived from aryl boroxines (Fig. 5a). In the presence of 3-ethynylanisole, CuI and Pd(PPh$_3$)$_2$Cl$_2$, polysubstituted 1,3-diene **45** was smoothly transformed into compound **67** via the Sonogashira cross-coupling reaction in excellent yield. Successively, several other Pd-catalyzed cross-coupling reactions with 1,3-diene **45** were also applicable. For instance, compound **45** could experience the Heck cross-coupling and Suzuki cross-coupling to access correspondingly products **68** and **69**, which bear three ester groups and could be easily delivered to diverse valuable functional groups if required.

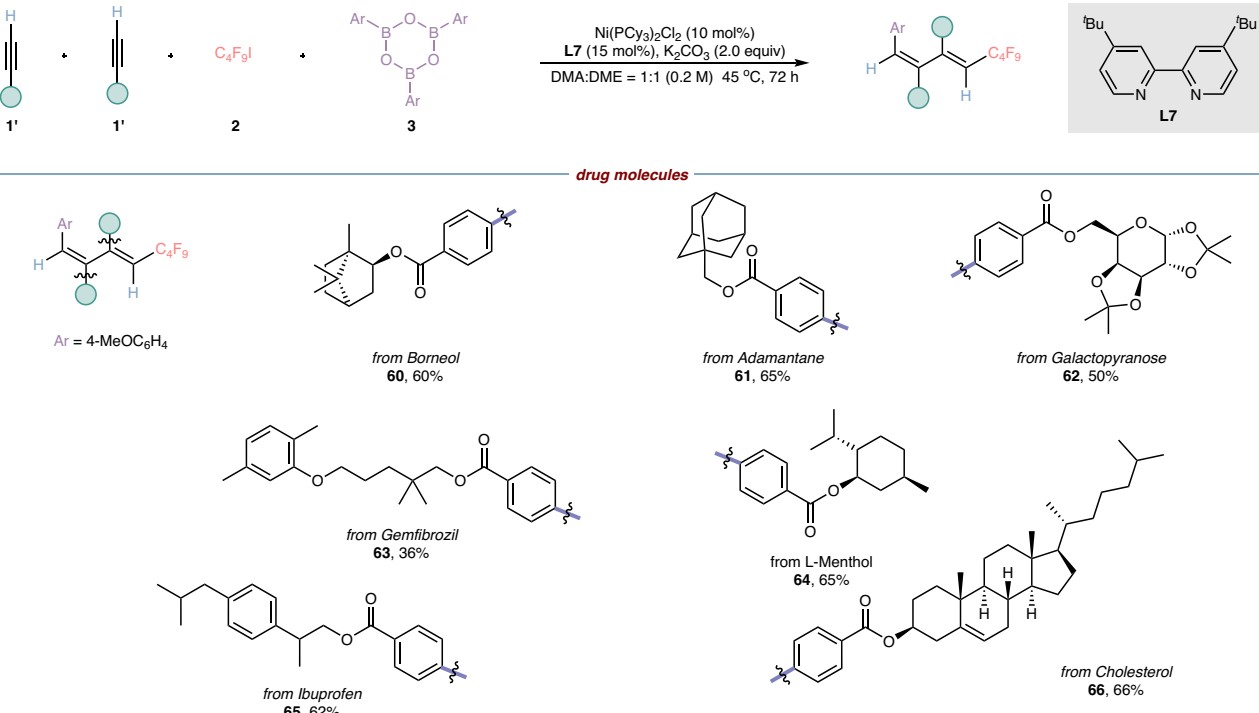

**Fig. 4 | Modification of natural products and drug molecules.** Reaction conditions: the reaction was carried out with **1'** (0.27 mmol), **2** (0.8 mmol), **3** (0.067 mmol), Ni(PCy₃)₂Cl₂ (10 mol%), dtbpy (15 mol%), K₂CO₃ (2.0 equiv) in 1 mL of mixed solvent (DMA:DME = 1:1) at 45 °C for 72 h, Unless otherwise stated, all desired polysubstituted 1,3-dienes were obtained with regioselectivity greater than 95:5, and the regioselectivity was detected by ¹H NMR analysis of desired products or GC analysis of desired products. DMA = N, N-dimethylacetamide, DME = 1,2-dimethoxyethane.

**Fig. 5 | Downstream applications and transformations. a** convert the reactive C-I bond derived from aryl boroxines. **b** transform the ester groups originated from accessible terminal alkynes. Reaction condition: [a]**45** (0.05 mmol), 3-ethynylanisole (3.0 equiv), PdCl₂(PPh₃)₂ (10 mol%), CuI (5 mol%) in Et₃N and THF at rt for 12 h. [b]**45** (0.05 mmol), methyl acrylate (5.0 equiv), Pd(ᵗBu₃P)₂ (10 mol%), and N, N-dicyclohexylmethylamine (2.0 equiv) in THF at 50 °C for 12 h. [c]**45** (0.05 mmol), (4-Ethoxycarbonylphenyl)boronic acid (1.5 equiv), Pd(PPh₃)₂Cl₂ (10 mol%), Na₂CO₃ (2.0 equiv) in THF and H₂O at 80 °C for 12 h. [d]**45** (0.05 mmol), Pd(OAc)₂ (5 mol%), Xantphos (10 mol%), Cs₂CO₃ (4.0 equiv) in 1,4-dioxane at 100 °C for 12 h. [e]**45** (0.05 mmol), DIBAL-H (10 equiv) in THF at −78 °C for 6 h. [f]**70** (0.05 mmol), PCC (2.4 equiv) in DCM at rt for 4 h. [g]**71** (0.05 mmol), MePPh₃Br (3.0 equiv), K₂CO₃ (4.0 equiv) in 1,4-dioxane at 80 °C for 12 h. [h]**45** (0.05 mmol), NaOH (20 equiv) in MeOH and THF at 80 °C for 8 h. DIBAL-H = diisobutylaluminium hydride, PCC = pyridinium chlorochromate.

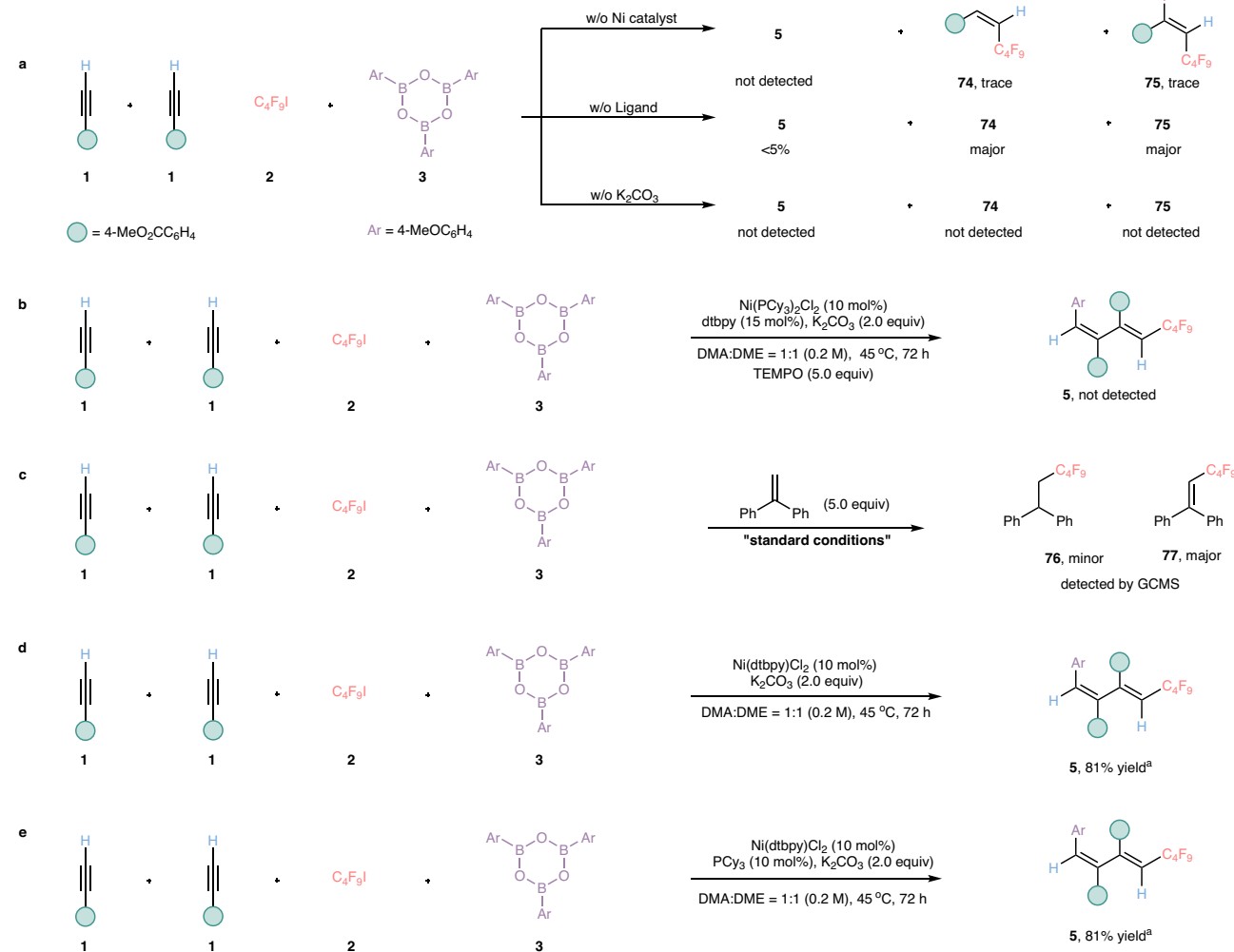

**Fig. 6 | Mechanism investigations. a** In the control experiments, metal catalysis, ligand, and base are essential for this catalytic cycle. **b** Radical inhibition reactions using TEMPO. **c** Radical trapping reactions using 1,1-Diphenylethylene. **d** Standard reaction condition using Ni(II) complex. dtbpy = 4,4′-di-tert-butyl-2,2′-dipyridyl. **e** To explore the role of PCy₃. ᵃ Yields were determined by gas chromatography (GC) using n-dodecane as the internal standard. TEMPO = 2,2,6,6-tetra-methylpiperidinyloxy, PCy₃ = tricyclohexyl phosphine, DMA = N, N-dimethylaceta-mide, DME = 1,2-dimethoxyethane.

Furthermore, a new C-N bond was forged from compound **45** and carbazole through Pd-catalyzed Buchwald-Hartwig cross-coupling reaction and compound **49** delivered in 92% yield (Fig. 5a). Subsequently, we were dedicated to transforming the ester groups originated from accessible terminal alkynes into diverse functional groups (Fig. 5b). For example, the ester group of **45** could groups (Fig. 5b, left). For example, the ester group of **45** could be successfully reduced to the corresponding benzyl alcohol **70** in the presence of DIBAL-H, and then alcohol **70** was transformed into aldehyde compound **71** through a one-step efficient oxidation reaction in excellent yield. Next, the Wittig-type reaction enabled the formation of styrene **72** in 90% yield from aldehyde **71** and MePPh₃Br in basic conditions. Last but not least, we also successfully access the 1,2,3,4-tetrasubstituted 1,3-diene carboxylic acid compounds **73** via hydrolysis of **45** under mild conditions with great outcome (Fig. 5b, right).

## Mechanism investigations

To shed light on the potential reaction pathways, several control experiments were carried out to explore the mechanism of this four-component reaction (Fig. 6). First of all, no expected product **5** was detected and only a trace amount of alkenes **74** and **75** were obtained in the lack of Ni catalyst, indicating that nickel catalyst is essential for

the generation of radicals and could be responsible for the process of syn-difunctionalization of the alkyne (Fig. 6a, top). Then, in the absence of bipyridine ligands, a dramatically decreased in the yield of compound **5** was observed and a substantial amount of fluoroalkylated alkene **74** and perfluoroalkylated alkenyl iodide **75** were detected, which revealed that the auxiliary effect of ligands on transition-metal catalysis significantly affects the process of trapping the newly generated fluoroalkylated alkenyl radical by Nickel species (Fig. 6a, middle). Crucially, the absence of base even inhibited the generation of radicals, this evidence illustrated that the transmetallation process was the initiation step of this process (Fig. 6a, bottom). Subsequently, when 2,2,6,6-tetramethy-1-pioerdinyloxy (TMEPO) was added to our catalytic system, the formation of compound **5** was significantly suppressed (Fig. 6b); when a stoichiometric amount of 1,1-diphenylethy-lene was added to the reaction, the perfluoroalkyl radical was successfully trapped, this result suggested that radical process could be involved in our transformation (Fig. 6c). Moreover, in recent years, ligand relay catalysis platform[63–67] has become a novel catalytic pattern in the construction of novel chemical bonds, the feasibility of ligand exchange could potentially improve the catalytic efficiency and get the satisfied results. As nitrogen ligand and phosphine ligand existed in our current reaction system, we guessed that whether our reaction

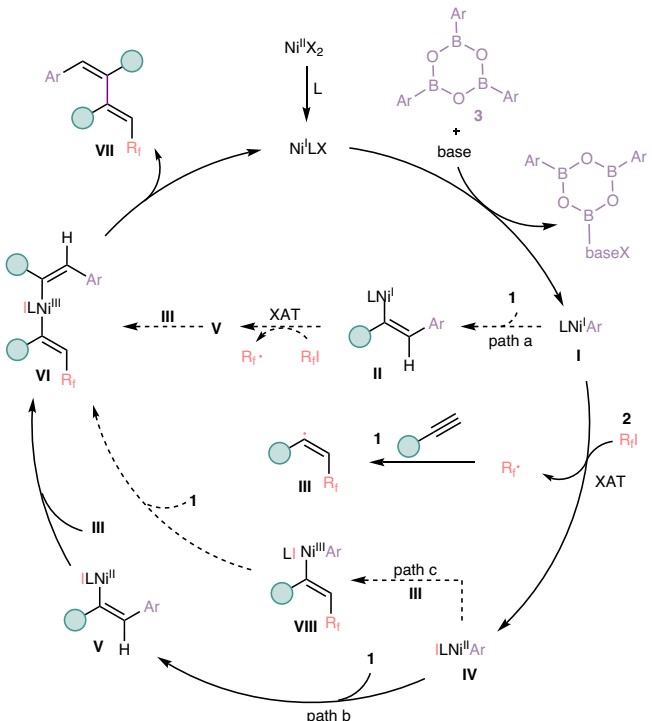

**Fig. 7 | Proposed reaction mechanism.** XAT = halogen atom transfer, L = ligand, X = halogen atom.

system probably underwent the process of ligand exchange to facilitate this unusual four-component reaction, yet further experiment indicated that dtbpy plays a leading role in our reaction system and excludes the ligands relay catalytic process (Fig. 6a, d and e).

On the basis of the previous literatures on Nickel-catalyzed difunctionalization of alkynes[37,38] and our observations, herein, three proposed reaction pathways of this four-component cascade reaction are described in Fig. 7. First of all, with the assistance of the base, aryl boroxine **3** and Ni(I) species undertake transmetallation process to render LNi$^I$Ar species **I** in situ. The species **I** could undergo regioselective syn-1,2-migratory insertion of alkyne **1** to afford alkenyl-Ni(I) intermediate **II**, which facilitates $R_f$-I to generate $R_f$-radical via XAT (halogen atom transfer) process to lead to alkenyl-Ni (II) intermediate **V**. Then, perfluoroalkyl radical undergoes radical addition to alkyne **1** to render the intermediate **III**, and the newly generated alkenyl radical **III** could be captured by intermediate **V** to form crucial Ni(III) species **VI** which followingly undergoes reductive elimination to access the desired fluoroalkylated polysubstituted 1,3-dienes **VII** (Fig. 7, Path a). In addition, with further studies on the transition-metal-catalyzed 1,2-difunctionalization of alkynes involving organoboron reagents[13] and transition-metal-catalyzed functionalization of π conjugated compounds[68-72], we propose another possible reaction pathway. After a similar transmetallation process, the LNi$^I$Ar species **I** mediates the XAT process of $R_f$-I **2** to deliver $R_f$-radical along with ILNi$^{II}$Ar (intermediate **IV**), and the intermediate **III** smoothly form from the same radical addition process as preceding Path a, simultaneously, another alkyne **1** undergoes critical migratory insertion into intermediate **IV** to deliver intermediate **V**, which traps the newly generated vinyl radical **III** to render the Ni(III) species **VI**, further reductive elimination renders the desired fluoroalkylated polysubstituted 1,3-dienes **VII** along with the regeneration of LNi$^I$ species, thus finishing the entire catalytic cycle (Fig. 7, Path b). Finally, according to Nevado's work[34,35,37,38], we also propose the third possible pathway as follows. In this pathway, intermediate **IV** generated from XAT process could recombine with the newly generated fluoroalkylated alkenyl radical **III** to render the

alkenyl-Ni(III) intermediate **VIII**, which occurs critical migratory insertion of alkyne **1** to afford Ni(III) species **VI** and subsequent reductive elimination completes the catalytic system and delivers the desired 1,2,3,4-tetrasubstituted 1,3-diene **VII** (Fig. 7, Path c).

To gain further insights on the possible mechanism of this reaction and verify the feasibility of these above-mentioned reaction pathways, theoretical investigation was performed at M06/6-311 + G(d, p)(SDD for Ni, K and I)/ SMD$_{DCM}$//M06/6-31 G(d)(SDD for Ni, K and I)/ SMD$_{DCM}$ level (Fig. 8). The combination of K$_2$CO$_3$, aryl boroxine **3**, and Ni(I)-iodide **CP16** results the generation of complex **CP1**, which is much stable comparing with simple Ni(I)-iodide **CP16**. Therefore, **CP1** was set to relative zero in calculated free energy profiles. Based on complex **CP1**, which undergoes a transmetallation process via transition state **TS2** to achieve aryl transfer with a free energy barrier of 22.5 kcal/mol. After the release of boron carbonate **CP3**, the Ni(I)-arylintermediate **CP4** is formed with an endothermic energy of 15.2 kcal/mol, regardless of the driving force from the crystallization process. Then, an XAT process between Ni(I)-aryl intermediate **CP4** and perfluorobutyl iodine **2** occurs via the transition state **TS5** affording a Ni(II)-iodide intermediate **CP7** with the release of perfluoroalkyl radical. The computational results showed that this process is quite fast with a free energy barrier of only 6.5 kcal/mol. The generated perfluoroalkyl radical could react with terminal alkyne **1** via radial addition transition state **TS8** to form a perfluoroalkylated alkenyl radical **CP9** with a free energy barrier of 9.9 kcal/mol. The formation of perfluoroalkylated alkenyl radical **CP9** is exergonic by 26.4 kcal/mol probably attributed to the generation of a new C-C covalent bond. Meanwhile, another terminal alkyne **1** could undergoes migratory insertion into Ni-C(aryl) bond via transition state **TS10** with a free energy barrier of 12.6 kcal/mol. The generation of Ni(II)-alkenyl intermediate **CP11** is also exergonic by 31.6 kcal/mol indicating an irreversible process. The combination of newly generated radical **CP9** and Ni(II)-alkenyl intermediate **CP11** results in a Ni(III)-bialkenyl intermediate **CP13** via radical trapping transition state **TS12** with a free energy barrier of 15.1 kcal/mol. The expected polysubstituted 1,3-diene product can be afforded through further reductive elimination from this Ni(III)-bialkenyl intermediate **CP13**. Computational results showed that the energy barrier of the step is only 7.4 kcal/mol. Finally, active species **CP1** can be regenerated with the ion exchange with K$_2$CO$_3$ and coordination of aryl boroxine (Fig. 8a). Hence, we successfully verify the feasibility of proposed Path b and reveal that this four-component reaction indeed achieves syn-difunctionalization and anti-difunctionalization of respective alkynes by a set of metal catalyst and ligand. At the same time, the origins of good regioselectivity without the use of directing group-containing[3,5,13] or electronically biased[3,5,13] alkynes were elaborately elucidated by subsequent calculation. (Please see the Supplementary Fig 28 for more details)

Furthermore, we have also investigated the competitive pathways (Path a and Path c) proposed in Fig. 7. As shown in Fig. 8b, when Ni(I)-aryl is formed, alkyne insertion could take place via transition state **TS18**. The calculated free energy barrier is 16.1 kcal/mol, which is much higher than that of XAT with **2**. Therefore, the formation of perfluoroalkyl radical **CP6** and Ni(II) intermediate **CP7** is the major pathway, thus the Path a is ruled out. As shown in Fig. 8c, the calculated free energy barrier of radical addition with perfluoroalkylated alkenyl radical **CP9** onto **CP7** via transition state **TS21** is 2.6 kcal/mol higher than that of alkyne insertion via transition state **TS10**. Hence, the generation of Ni(III)-alkenyl-aryl intermediate **CP22** is unfavorable, which could derive to the formation of polysubstituted alkenes and return to the well-known three-component reaction, this result also led us to exclude the Path c. Notably, in our theoretical calculation, we found that the rate of alkyne insertion is faster than that of radical addition, so 1,2,3,4-tetrasubstituted 1,3-dienes could be found as a primary product rather than trisubstituted alkenes, thus well explaining that the success of modulating the reaction from conventional

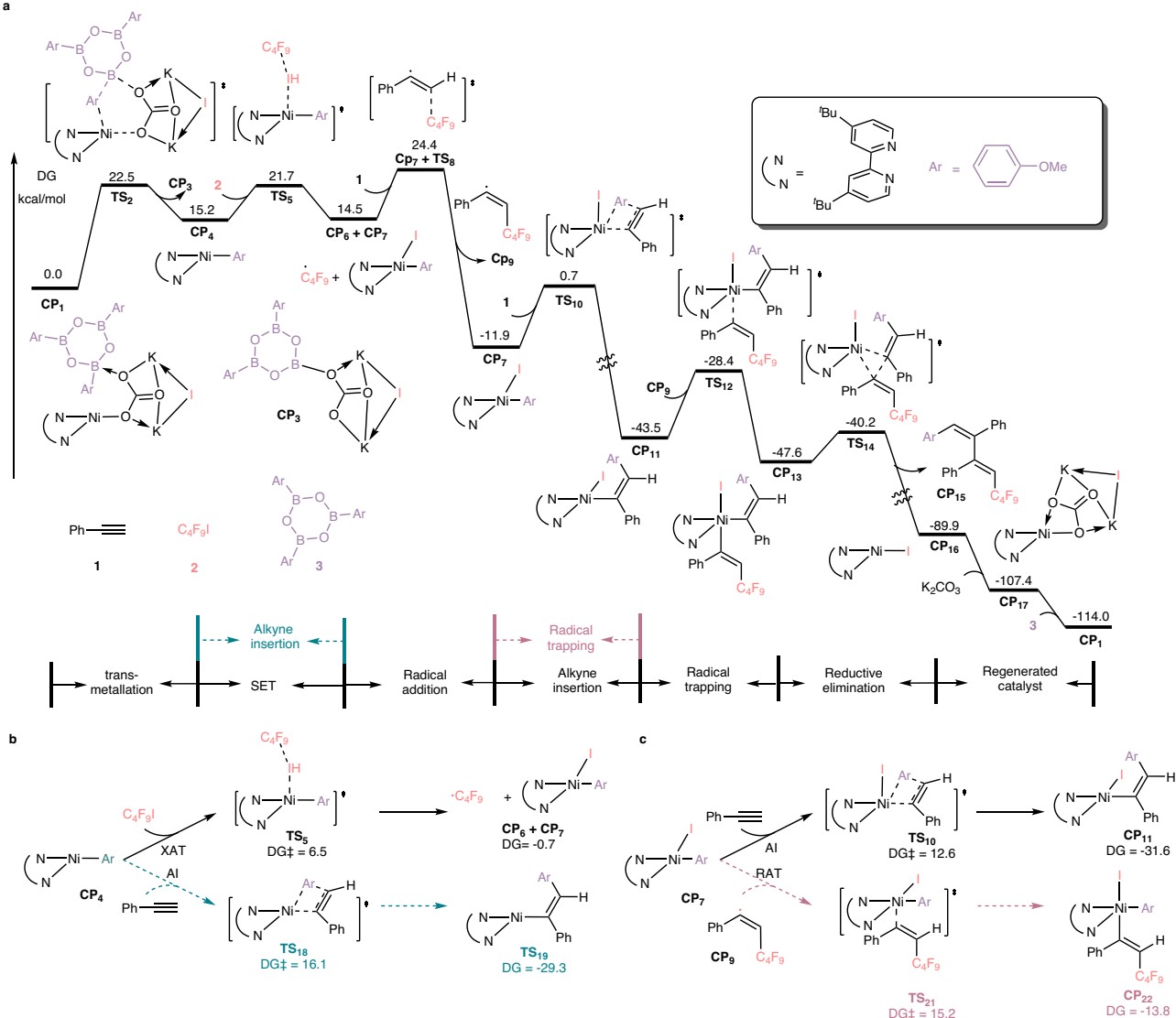

**Fig. 8 | The calculated results at M06/6-311 + G(d, p)(SDD for Ni, K and I)/ SMD_DCM//M06/6-31 G(d) (SDD for Ni, K and I)/SMD_DCM. a** The free energy profile for the Nickel-catalyzed four-component cascade difunctionalization of alkynes. **b** The comparing pathways between the XAT and alkyne insertion for the Ni(I)-aryl species. **c** The comparing pathways between the radical trapping and alkyne insertion for the Ni(II)-aryl species. AI = alkyne insertion, RAT = radical trapping, XAT = halogen atom transfer.

three-component reaction transforms into unusual and challenging four-component reaction in our catalytic system.

In summary, we report a Nickel-catalyzed four-component reaction involving two terminal alkynes, aryl boroxine, and perfluoroalkyl iodides, which provide a simple and flexible platform to access polysubstituted 1,3-dienes via unusual cascade difunctionalization. Herein, we successfully combine the carbometallation/cross-coupling platform and radical addition/cross-coupling platform in one operation, which utilizes cheap nickel catalyst and accessible starting materials to render the more complicated fluorinated 1,2,3,4-tetrasubstituted conjugated dienes and proposes the possible reaction mechanism under our catalytic system with both anti-difunctionalization and syn-difunctionalization of two terminal alkynes. This protocol features mild conditions, wide substrate compatibility, simple execution, excellent chemoselectivity, good regioselectivity, and exclusive stereoselectivity. The late-stage transformations demonstrate the synthetic application of our current strategy and offer a facile route to access an array of versatile 1,2,3,4-tetrasubstituted 1,3-dienes. Mechanistic experiments reveal the possible mechanism of this

cascade difunctionalization reaction and density functional theory (DFT) calculations further verify the feasibility of the proposed mechanism as well as explain why our reaction tends to an unusual four-component reaction rather than the well-established three-component reaction from similar starting materials. Further application and the selectivity of different alkynes owing to greater challenges in our reaction system will be explored by our laboratory in the future.

## Methods
### General procedure for the synthesis of polysubstituted 1,3-dienes

To an oven-dried 10 mL Young's Tube vial equipped with a magnetic stir bar was added L7 (0.03 mmol, 8.0 mg, 15 mol%), Ni(PCy₃)₂Cl₂ (0.02 mmol, 13.8 mg, 10 mol%), terminal alkynes (0.27 mmol, 1.35 equiv), boroxine (0.067 mmol, 27 mg), K₂CO₃ (0.4 mmol, 56 mg, 2.0 equiv), The vial was introduced in an argon-filled atmosphere, then perfluoroalkyl iodides (0.8 mmol, 4.0 equiv), anhydrous DMA (0.5 mL, 0.2 M) and anhydrous DME (0.5 mL, 0.2 M) were added. Next, the reaction mixture was stirred at 45 °C (or 50 °C) in an oil bath at

660 rpm for 72 h. After the reaction was completed, the reaction mixture was extracted with EtOAc, and the combined organic layers were dried over $Na_2SO_4$, filtered, and concentrated under reduced pressure. The product was purified by column chromatography over silica gel for each substrate.

## Data availability

Data relating to the materials and methods, optimization studies, experimental procedures, NMR spectra, and mass spectrometry are available in the Supplementary Information. Source data are provided in this paper. Crystallographic data for the structures reported in this Article have been deposited at the Cambridge Crystallographic Data Center, under deposition numbers 2289202 (**58**) and 2289200 (**59**). Copies of the data can be obtained free of charge via https://www.ccdc.cam.ac.uk/structures/. Data can also be obtained from the corresponding author upon request. Source data are provided with this paper.

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

## Acknowledgements

Financial support from the National Key R&D Program of China (2023YFF0723900 to Q.S.), National Natural Science Foundation of China (21931013 and 22271105 to Q.S.), Natural Science Foundation of Fujian Province (2022J02009 to Q.S.) and Open Research Fund of School of Chemistry and Chemical Engineering, Henan Normal University (to Q.S.) are gratefully acknowledged.

## Author contributions

Q.S. designed and directed the project. S.C. performed the experiments and developed the reactions. J.X., W.L., M.Y., X.M., and K.Y. helped collect some experimental data; Y.L. and S.L. directed the DFT calculations, and Y.W. conducted the DFT calculations. Q.S., Y.L., and S.C. wrote the manuscript. All authors discussed the results and commented on the manuscript.

## Competing interests

The authors declare no competing interests.
