## [Peer Review File · Nature Communications]

Chemo-, regio- and stereoselective access to polysubstituted 1,3-dienes via Nickel-catalyzed four-component reactionsEditorial Note: Parts of this Peer Review File have been redacted as indicated to remove third-party material where no permission to publish could be obtained.

REVIEWER COMMENTS

Reviewer #1 (Remarks to the Author):

The manuscript from Song, Lan, and coworkers describes a chemo-, regio- and stereoselective Ni-catalyzed four-component reaction of two terminal alkynes, aryl boroxines, and perfluoroalkyl iodides, providing rapid access to polysubstituted 1,3-dienes. The synthetic utility of our protocol is further demonstrated by the late-stage transformations. In addition, the related reaction mechanism was confirmed by experiments and DFT calculation. I am enthusiastic about this article, and I feel that it will be suitable for publication in Nat. Commun. after the minor points below have been addressed.

Specific Comments:

- 1) A significant drawback of this transformation is the use of largely excessive of alkynes (2*4 equiv.) and perfluoroalkyl iodides (11 equiv). Information about the detailed analysis of the byproducts and the isomeric products are suggested to be added. In conditions optimization Table 2, the authors are suggested to add the informations of Z/E, rr (ratio of 5:5').
- 2) In Figure 1, alkynes used were drawn as internal alkynes. However, in Fig. 2, only examples of terminal alkynes were illustrated. Information about using internal alkynes should be mentioned. Otherwise, please change the alkyne structures in Fig. 1. Currently, aryl-substituted terminal alkynes were used. How about other alkyl-substituted terminal alkynes?
- 3) In terms of the substrate scope of perfluoroalkyl iodides, could non-fluoroalkyl iodide or CF₃I be suitable substrate?
- 4) Could aryl halides be used instead of aryl boroxines under reductive conditions?

Reviewer #2 (Remarks to the Author):

This work by Song et al presents a chemo-, regio- and stereoselective synthesis of polysubstituted 1,3-dienes through a nickel-catalyzed four-component reaction. The process features two terminal alkynes, aryl boroxines, and perfluoroalkyl iodides as reaction partners. Several precedents have been published describing the internal homocoupling of alkynes after the addition of a C-based functionality stemming from aryl boronic acid derivatives. However, no examples of a tetracomponent reaction have been reported up to date, which certainly adds an element of novelty to this contribution. The following references should be added to the introduction and the discussion of the precedents

1. Ligand-Controlled Cross-Dimerization and -Trimerization of Alkynes under Nickel Catalysis <https://doi.org/10.1002/adsc.200800399>
2. Nickel-catalysed hydroarylation of alkynes using arylboron compounds: selective synthesis of multisubstituted arylalkenes and arylidienes: <https://doi.org/10.1039/B107866F>
3. Synthesis of gem-Difluorinated 1,4-Dienes via Nickel-Catalyzed Three-Component Coupling of (Trifluoromethyl)alkenes, Alkynes, and Organoboronic Acids | Organic Letters (acs.org)

The reaction scope is limited to the incorporation of two identical alkynes. No example has

been carried out using two different alkynes, which even if in a mixture, would provide additional (also interesting) products. It would be appealing to explore how different electronics on the substituents of the triple bond (e-donating vs. e-withdrawing) affect the selectivity and the overall reaction outcome. The authors should also explore and comment on the use of alkyl-substituted alkynes, currently not featured in the paper.

The major limitation of the chemistry is the fact that a single radical precursor type, namely perfluoroalkyl-iodides seem to be compatible with the optimized protocol. Other more electron deficient (α -bromo ketones, α -bromo esters, α -bromo- α -difluoro esters) and electron rich (tBul) precursors should also be explored in this context.

The authors have further functionalized the products via C-C bond formation at the aryl moiety. However, it would be important to demonstrate some derivatization of the diene scaffold itself, in the form of hydrogenation, epoxydations, conjugated additions, diels-alder reactions etc. to name a few.

The proposed mechanism is in line with the control experiments carried out. Even if they are not very indicative, the DFT calculations and the bibliography endorses the mechanism.

Thus, after addressing the abovementioned points to truly showcase the synthetic potential of the method, publication of this manuscript in Nat. Commun. can be recommended.

Reviewer #3 (Remarks to the Author):

Song and co-workers reported a novel Nickel-catalyzed four-component reaction to access densely substituted 1,3-dienes using two terminal alkynes, aryl boroxines, and perfluoroalkyl iodides. The excellent chemoselectivity, good regioselectivity and exclusive stereoselectivity was observed by combine the carbometallation/cross-coupling platform and radical addition/cross-coupling platform in one operation. Control experiments reveal the plausible reaction mechanism, and using DFT calculations, the authors proposed a reasonable mechanism that account the formation of this unusual four-component reaction. I would like to recomend to publish after addressing the following issues.

1.Can ArBpin been applied with water in the low-polar solvent (like toluene)?

2.The type of salts has a significant impact on the activity of the reaction. I agree with the authors that the salts assisted transfer of aromatic groups. However, the authors' proposed mechanism does not explain the necessity of K₂CO₃. In addition, author's mechanism experiments and previous computational studies on iodides show that base can facilitate the SET process. This should be made clear to the reader.

3.Related literatures should be added in the introduction part: J. Org. Chem. 2023, 88, 14115–14130; Eur. J. Org. Chem. DOI:10.1002/ejoc.202201422; J. Org. Chem. DOI:10.1021/acs.joc.3c02667

Point-by-point response to reviewers' comments

REVIEWER COMMENTS

Reviewer #1 (Remarks to the Author):

The manuscript from Song, Lan, and coworkers describes a chemo-, regio- and stereoselective Ni-catalyzed four-component reaction of two terminal alkynes, aryl boroxines, and perfluoroalkyl iodides, providing rapid access to polysubstituted 1,3-dienes. The synthetic utility of our protocol is further demonstrated by the late-stage transformations. In addition, the related reaction mechanism was confirmed by experiments and DFT calculation. I am enthusiastic about this article, and I feel that it will be suitable for publication in Nat. Commun. after the minor points below have been addressed.

Response: We sincerely thank this reviewer for the favorable comments on our work. We really appreciate it. And we have revised the manuscript and Supporting Information according to the comments. We will answer your questions in detail shown below:

Specific Comments:

1) A significant drawback of this transformation is the use of largely excessive of alkynes (2*4 equiv.) and perfluoroalkyl iodides (11 equiv). Information about the detailed analysis of the byproducts and the isomeric products are suggested to be added. In conditions optimization Table 2, the authors are suggested to add the informations of Z/E, rr (ratio of 5:5').

Response: We deeply thank this reviewer for the constructive and inspiring suggestions. According to the previous literature (*Angew. Chem. Int. Ed.* **2013**, *52*, 13745-13750.), aryl boroxines are trimers of aryl boronic acid obtained by dehydrative condensation and the use of aryl boroxines as aryl sources generally need to reduce the amount to one third. In our catalytic system, the reaction was 0.2 mmol scale and we selected aryl boroxines as 1.0 equiv, so the use of aryl source was 0.2 mmol (namely aryl boroxine were 0.667 mmol). Meanwhile, the use of alkyne was 2.7 equivalent (0.54 mmol) in total and the use of perfluoroalkyl iodides were 4.0 equivalent (0.8 mmol). These are our negligence, we have corrected the relevant amount of alkynes to 0.27 mmol (0.27 mmol*2 in total), please see our revised manuscript.

Information about the detailed analysis of the byproducts and the isomeric products have been added to the Supporting Information. Please see our revised Supporting Information on page 4 for details.

The informations of Z/E, rr (ratio of 5:5') of our desired products have been added in Table 2. Please see our revised manuscript for details.

2) In Figure 1, alkynes used were drawn as internal alkynes. However, in Fig. 2, only examples of terminal alkynes were illustrated. Information about using internal alkynes

should be mentioned. Otherwise, please change the alkyne structures in Fig. 1. Currently, aryl-substituted terminal alkynes were used. How about other alkyl-substituted terminal alkynes?

Response: We sincerely thank this reviewer for the inspiring and constructive suggestions, we really appreciate it. By the survey of literature (*ACS Catal.* **2021**, *11*, 7513–7551, *Nat. Commun.* **2018**, *9*, 4543-4550, *J. Am. Chem. Soc.* **2023**, *145*, 18722–18730), we have found that transition-metal-catalyzed 1,2-difunctionalization of alkynes involving organoboron reagents could be divided into radical addition/cross-coupling tactic and carbometallation/cross-coupling platform. Please take a look downwards.

[Redacted]

ACS Catal. **2021**, *11*, 7513–7551.

On the one hand, radical addition/cross-coupling tactic is usually applied to terminal alkynes and the successful transformations of such reaction pattern have been well studied so far (*ACS Catal.* **2021**, *11*, 7513–7551), yet the success of internal alkyne is very rare (*Top Catal.* **2017**, *60*, 545-553; *ACS Catal.* **2016**, *6*, 3452–3456).

On the other hand, carbometallation/cross-coupling platform involving organoboron reagents is usually applied to internal alkynes (*ACS Catal.* **2021**, *11*, 7513-7551, *Nat. Commun.* **2018**, *9*, 4543-4550, *J. Am. Chem. Soc.* **2023**, *145*, 18722-18730), in general, electronically biased or directing-auxiliary-containing alkynes have been used to control regioselectivity. Moreover, the success of such reaction pattern involving internal alkynes have been well studied up to date (*ACS Catal.* **2021**, *11*, 7513–7551), yet the successful transformation of terminal alkyne is few.

Hence, in our Figure. 1, we use the alkyne structure to include both terminal alkynes and internal alkynes. The blue sphere represents the group with lower steric hindrance, which is conducive to radical addition first.

[Redacted]

ACS Catal. **2021**, *11*, 7513–7551.

As a result, we thought that alkyne structures are terminal alkyne or internal alkyne in respective difunctionalization patterns, which promote us to use two different spheres to represent the substituents on the both sides of the alkynes.

In terms of “Currently, aryl-substituted terminal alkynes were used. How about other alkyl-substituted terminal alkynes?”, actually before the submission, a variety of alkyl-substituted terminal alkynes have been examined and no desired products were formed, and we have validated the reaction results several times and every time obtained the same results. And the relevant reaction outcomes have been indicated in supporting information on page 70. Please see our revised supporting information for more details.

3) In terms of the substrate scope of perfluoroalkyl iodides, could non-fluoroalkyl iodide or CF₃I be suitable substrate?

Response: We deeply thank this reviewer for raising this issue, it is really inspiring and constructive. Actually, before the submission, we had examined a variety of non-fluoroalkyl iodides and found that no desired products were obtained in all cases. In addition, when we used the CF₃I as radical precursor in our current reaction system, we merely detected a trace amount of product formed. So, non-fluoroalkyl iodide or CF₃I are not suitable substrates for our current reaction system. The corresponding modifications have been made in the supporting information. Please see our revised supporting information on page 70 for more details.

4) Could aryl halides be used instead of aryl boroxines under reductive conditions?

Response: We deeply thank this reviewer for the inspiring comments. That's a good suggestion, per the request, we have examined aryl halides (such as aryl bromide and aryl iodide) as substrates under reductive conditions, however, it was unable to afford the desired products and the 1,3-enyne compounds were detected as major products by GCMS. The relevant experimental results are showed as follows.

Reviewer #2 (Remarks to the Author):

This work by Song et al presents a chemo-, regio- and stereoselective synthesis of polysubstituted 1,3-dienes through a nickel-catalyzed four-component reaction. The process features two terminal alkynes, aryl boroxines, and perfluoroalkyl iodides as reaction partners.

Several precedents have been published describing the internal homocoupling of alkynes after the addition of a C-based functionality stemming from aryl boronic acid derivatives. However, no examples of a tetracomponent reaction have been reported up to date, which certainly adds an element of novelty to this contribution. The following references should be added to the introduction and the discussion of the precedents

1. Ligand-Controlled Cross-Dimerization and -Trimerization of Alkynes under Nickel Catalysis <https://doi.org/10.1002/adsc.200800399>
2. Nickel-catalysed hydroarylation of alkynes using arylboron compounds: selective synthesis of multisubstituted arylalkenes and arylidienes: <https://doi.org/10.1039/B107866F>
3. Synthesis of gem-Difluorinated 1,4-Dienes via Nickel-Catalyzed Three-Component Coupling of (Trifluoromethyl)alkenes, Alkynes, and Organoboronic Acids | Organic Letters ([acs.org](https://doi.org/10.1021/acs.orglett.2c01111))

Response: We sincerely thank this reviewer for recommending these highly-related reviews which are very important and meaningful. Per the request, the corresponding references (*Adv. Synth. Catal.* **2008**, *350*, 2274-2278; *Chem. Commun.* **2001**, 2001 2688-2689; *Org. Lett.* **2023**, *25*, 1748-1753) have been cited in the revised manuscript (Please see Ref. 68, Ref. 69 and Ref. 21).

The reaction scope is limited to the incorporation of two identical alkynes. No example has been carried out using two different alkynes, which even if in a mixture, would

provide additional (also interesting) products. It would be appealing to explore how different electronics on the substituents of the triple bond (e-donating vs. e-withdrawing) affect the selectivity and the overall reaction outcome. The authors should also explore and comment on the use of alkyl-substituted alkynes, currently not featured in the paper.

Response: We sincerely thank this reviewer for pointing it out. Actually, before the submission, we have examined two different alkynes in our current catalytic system several times and no good results were obtained. No matter that they are neutral groups, donating groups or withdrawing groups on the *para*-position of the aromatic alkynes, only a mixture was obtained (formed four products), which may be due to the poor chemoselectivity of radical addition and migratory insertion process. Moreover, we assumed that the electronic properties of the aryl ring would greatly affect the corresponding chemoselectivity and help us to solve the problem and get satisfactory results. So we tried to the use of largely excessive of electron-rich (such as 4-methoxyphenylacetylene), electron-deficient (such as methyl 4-ethynylbenzoate) and electron neutral aromatic terminal alkynes (such as phenylacetylene), however, the desired results still could not be achieved. Please take a look downwards.

Two different alkynes were used in our currently reaction system.

The use of largely excessive of one alkyne

Therefore, different electronic properties of the substituents on the aromatic alkynes couldn't affect the selectivity, even with largely excessive of one alkyne. Furthermore, we also explored the alkyl-substituted alkynes as substrates in our current catalytic

system, yet they were unable to render the expected products too. The experimental results have been added to the revised supporting information. Please see our revised supporting information on page 68 for more details.

The major limitation of the chemistry is the fact that a single radical precursor type, namely perfluoroalkyl-iodides seem to be compatible with the optimized protocol. Other more electron deficient (α -bromo ketones, α -bromo esters, α -bromo- α -difluoro esters) and electron rich (^tBuI) precursors should also be explored in this context.

Response: We deeply thank this reviewer for the constructive and inspiring suggestions. Actually, before the submission, we have examined a variety of non-fluoroalkyl iodide but couldn't obtain the expected results. A variety of common radical precursors such as α -bromo ketones, α -bromo esters, α -bromo- α -difluoro esters and ^tBuI have been examined in our reaction system, but we could not detect the expected products formed. We have added the results in our revised supporting information accordingly. Please see our revised supporting information on page 70 for more details.

The authors have further functionalized the products via C-C bond formation at the aryl moiety. However, it would be important to demonstrate some derivatization of the diene scaffold itself, in the form of hydrogenation, epoxydations, conjugated additions, diels-alder reactions etc. to name a few.

Response: We deeply thank this reviewer for the constructive and inspiring suggestions. Actually, before the submission, we had examined a variety of derivatizations of the corresponding diene scaffolds. Unfortunately, neither hydrogenation, epoxydations, conjugated additions, nor diels-alder reactions of our polysubstituted 1,3-dienes could get the satisfied results and a large amount of the raw material residue was detected. It might be attribute to the steric hindrance of our polysubstituted 1,3-dienes compounds, which is not conducive to the reaction proceeding. Please look the following content.

According to previous literature (*Chin. Chem. Lett.*, **2022**, *33*, 4074–4078), we assume that the electronic effect of perfluoroalkyl group on 1,3-dienes compounds will change its chemical properties, which make it is difficult to be reduced to carbon-carbon single bonds by hydrogen. In conclusion, we have not successfully achieved the

hydrogenation reaction of our polysubstituted 1,3-diene so far.

Additionally, epoxydations, conjugated additions, Diels-Alder reactions of our polysubstituted 1,3-dienes had also been examined for several times and obtained the same results (only a large amount of the raw materials was detected). These results may be due to the steric hindrance and the electronic effect of our polysubstituted 1,3-dienes, which are not conducive to the reaction proceeding. Please look the following content.

The proposed mechanism is in line with the control experiments carried out. Even if they are not very indicative, the DFT calculations and the bibliography endorses the mechanism.

Response: We deeply thank this reviewer for the valuable comments. Actually, before the submission, in order to better illustrate the proposed mechanism of our reaction, we

carried out other control experiments as followed, even though they couldn't provide any indicative information. Please take a look downwards.

The unique reaction pattern and structurally diversified polysubstituted 1,3-dienes compounds promote us to verify the mechanism of the reaction and explain the cause for the formation of this unusual four-component reaction instead of classical three-component difunctionalization reaction through density functional theory (DFT) calculations. According to the previous literature reports and current control experiments, we have proposed three possible mechanisms and verify the feasibility of these reaction pathways via DFT calculations. Finally, we found that path b proposed in our manuscript is the most reasonable reaction pathway for this unusual four-component reaction.

Thus, after addressing the abovementioned points to truly showcase the synthetic potential of the method, publication of this manuscript in Nat. Commun. can be recommended.

Our response: We sincerely thank this reviewer for the valuable and positive comments on our manuscript, we really appreciate it. We have answered all of the questions raised by this reviewer one by one in detail. Thank you very much for your support.

Reviewer #3 (Remarks to the Author):

Song and co-workers reported a novel Nickel-catalyzed four-component reaction to access densely substituted 1,3-dienes using two terminal alkynes, aryl boroxines, and perfluoroalkyl iodides. The excellent chemoselectivity, good regioselectivity and

exclusive stereoselectivity was observed by combine the carbometallation/cross-coupling platform and radical addition/cross-coupling platform in one operation. Control experiments reveal the plausible reaction mechanism, and using DFT calculations, the authors proposed a reasonable mechanism that account the formation of this unusual four-component reaction. I would like to recommend to publish after addressing the following issues.

Response: We deeply thank the reviewer for his/her inspiring and favorable comments on our manuscript, we really appreciate it. And we have revised the manuscript according to the comments. We will answer your questions in detail shown below:

1. Can ArBpin been applied with water in the low-polar solvent (like toluene)?

Response: We deeply thank this reviewer for the valuable suggestion. In our current reaction system, the use of ArBpin with water in the toluene couldn't render the desired polysubstituted 1,3-dienes, to our surprised, we detected the large amount of other products (shown below). We repeated the reaction several times and every time the same results were obtained.

The type of salts has a significant impact on the activity of the reaction. I agree with the authors that the salts assisted transfer of aromatic groups. However, the authors' proposed mechanism does not explain the necessity of K₂CO₃. In addition, author's mechanism experiments and previous computational studies on iodides show that base can facilitate the SET process. This should be made clear to the reader.

Response: We deeply thank this reviewer for the constructive and inspiring suggestions. And many thanks for the agreement with our proposed role for the base K₂CO₃. Actually, as shown in Table 2, the absence of base will greatly hamper the reaction progress and suppress the reaction, which suggests that it is crucial for the success of this four-component reaction. In order to give a clearer glance at the role of base, we tried to find out the directly transmetalation process via a very high free energy barrier of 50.8 kcal/mol in the absence of base (Figure 5 in the supporting information). Furthermore, the formation of Ni(I)-arylintermidiate CP4 would be endothermic process with a free energy of 48.4 kcal/mol, causing to the following process hard to happen. Comparing

with the base assisted transfer of aromatic group, it's clear to find out that the base is necessary in the experimental condition. In addition, it should be mentioned that the process between the Ni(I) species and perfluoroalkyl iodide should be an XAT (halogen atom transfer) process with a free energy of 9.9 kcal/mol instead of an SET (single electron transfer) process. Sorry for the negligence. We have corrected the expression for main article and Figure 8 in revised manuscript.

Supplementary Figure 5. The free energy profile for the transmetalation process in the absence of base.

Table 1

Legend: $\text{Ar} = 4\text{-MeOC}_6\text{H}_4$

18 \leftarrow	- \leftarrow	L7 \leftarrow	K ₂ CO ₃ \leftarrow	DMA:DME (1:1) \leftarrow	nd \leftarrow
19 \leftarrow	Ni(PCy ₃) ₂ Cl ₂ \leftarrow	- \leftarrow	K ₂ CO ₃ \leftarrow	DMA:DME (1:1) \leftarrow	< 5 \leftarrow
20 \leftarrow	Ni(PCy ₃) ₂ Cl ₂ \leftarrow	L7 \leftarrow	- \leftarrow	DMA:DME (1:1) \leftarrow	nd \leftarrow

Ligands L1, L7, L8, L9, L10, L11 are shown below the table.

Control experiment

Proposed reaction mechanism

Figure 7. Proposed reaction mechanism.

3. Related literatures should be added in the introduction part: *J. Org. Chem.* 2023, 88, 14115–14130; *Eur. J. Org. Chem.* DOI:10.1002/ejoc.202201422; *J. Org. Chem.* DOI:10.1021/acs.joc.3c02667

Response: We sincerely thank this reviewer for recommending these highly related articles that are very important and meaningful. The corresponding references (*J. Org. Chem.* 2023, 88, 14115–14130; *Eur. J. Org. Chem.* DOI:10.1002/ejoc.202201422; *J. Org. Chem.* 2024, 89, 4484–4495) have been cited in the revised manuscript (Please see Ref. 70, Ref. 71 and Ref. 72).

REVIEWERS' COMMENTS

Reviewer #1 (Remarks to the Author):

The Authors have addressed all of reviewers' concerns with the original manuscript. The revised manuscript is now ready for publication.

Reviewer #3 (Remarks to the Author):

Song and co-workers reported a novel Nickel-catalyzed four-component reaction to access densely substituted 1,3-dienes using two terminal alkynes, aryl boroxines, and perfluoroalkyl iodides.

Control experiments reveal the plausible reaction mechanism, and using DFT calculations, the authors proposed a reasonable mechanism that account the formation of this unusual four-component reaction. The author's response completely solves my question, I can recommend this work.

REVIEWER COMMENTS

Reviewer #1 (Remarks to the Author):

The Authors have addressed all of reviewers' concerns with the original manuscript. The revised manuscript is now ready for publication.

Our Response: We sincerely thank this reviewer for his/her favorable comments on our manuscript, we really appreciate it!

Reviewer #3 (Remarks to the Author):

Song and co-workers reported a novel Nickel-catalyzed four-component reaction to access densely substituted 1,3-dienes using two terminal alkynes, aryl boroxines, and perfluoroalkyl iodides.

Control experiments reveal the plausible reaction mechanism, and using DFT calculations, the authors proposed a reasonable mechanism that account the formation of this unusual four-component reaction. The author's response completely solves my question, I can recommend this work.

Our Response: We sincerely thank this reviewer for the favorable comments on our work. We really appreciate it.